# GANOCS: Domain Adaptation of Normalized Object Coordinate Prediction Using Generative Adversarial Training

**Bahaa Aldeeb, Sahith Chada, Karthik Desingh**
Department of Computer Science and Engineering
University of Minnesota, United States
{baldeeb, chada022, kdesingh}@umn.edu

**Abstract:** Estimating 2D-3D correspondences has proven to be a very useful tool for category-level pose and scale estimation and robot manipulation tasks; however, it is hindered by the difficulty of obtaining 3D object models and labels. Simulation reduces the burden of labeling but introduces a gap between the training and operational domains. We introduce a novel architecture for integrating cross-domain data in the training of NOCS predictors (a form of 2D-3D correspondences). We leverage Generative-Adversarial-Networks (GANs) to avoid the need for burdensome real-data labeling by using a domain-agnostic discriminator as a supervisor. This work presents results demonstrating the potential of our method.

**Keywords:** Domain Adaptation, 2D-3D Correspondences, 3D Perception, Semi-supervised Learning

## 1 Introduction

Perceiving 3D object properties such as poses, shapes, and sizes from 2D data can benefit embodied AI applications such as navigation [1], manipulation [2, 3], and pose estimation [4]. While unsupervised training and large datasets helped advance classification, segmentation, and detection tasks [5, 6], 3D perception did not reap equal benefits and remains a topic of research because of the difficulty of acquiring such large datasets [7].

Obtaining dense correspondences between seen object images and their 3D model (2D-3D correspondences) is a form of 3D perception that is useful for category-level pose estimation [4, 8, 7] as well as manipulation [2, 3]. The Normalized Object Coordinate System (NOCS) was presented by Wang et al. [4] for deriving 2D-3D correspondence representations. Representing an object using NOCS involves scaling 3D object models into a unit cube, consistently orienting object instances within each category, and expressing $(x, y, z)$ points on the surface of those normalized objects as $(r, g, b)$ colors. Projection of those colors onto an image plane produces NOCS maps that machine learning models can be trained to predict from visual data. NOCS map predictors [8, 7, 9] have shown state-of-the-art results on category-level pose and scale estimation. NOCS representations have been used for manipulation policy learning [2, 3] and similar dense representations have been used to extract keypoints to make manipulation policy learning more efficient [10, 11]. The ability to predict NOCS from RGB images also makes it a beneficial tool for 3D perception and localization of distant objects in domains that lack dense depth data such as autonomous driving [1].

The demand for 3D data labeling is a key challenge for 2D-3D correspondence learning [12, 4]. Getting a ground truth NOCS map of an object in an image requires labeling the image using the 3D object models and their 6D pose [4]. While acquiring such labels is relatively trivial in simulation, training using only simulated data is insufficient if the model is to be used on real data due to the gap between those operational domains. These issues motivated the development of domain adaption [7, 9] and self-supervised methods [12, 13]. While those methods produce impressive results, they rely on accurate depth for training. We aim to mitigate the difficulties of real-data labeling and address the sim-to-real domain shift when training an RGB-conditioned NOCS detector.

7th Conference on Robot Learning (CoRL 2023), Atlanta, USA.

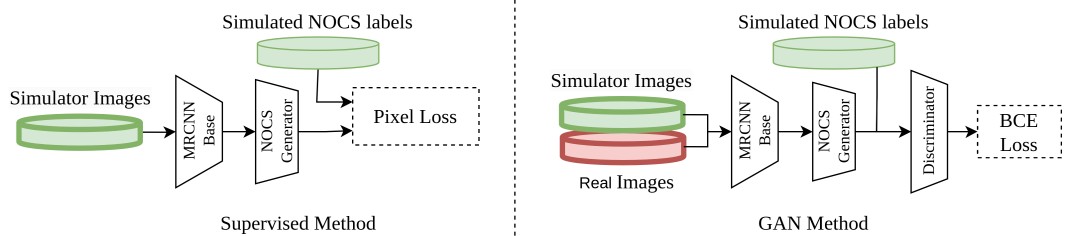

Figure 1: This diagram illustrates the difference between supervised NOCS training (left) and our proposed adversarial training (right). Since NOCS labels are available only for simulated data, an explicitly supervised NOCS generator (left) could only train on the corresponding simulated images. Our proposed GANOCS generator can get indirect supervision from out-of-distribution data from real images by leveraging a domain-agnostic discriminator (right). We train the discriminator to be agnostic to the input domain by conditioning it only on the NOCS maps. Our generator still benefits from supervised pixel-wise loss when available.

We present GANOCS, as a way to train a NOCS predictor using unlabeled real-world data by leveraging Generative Adversarial Networks (GANs). Our primary insight is that ground truth NOCS maps of real and simulated images are very similar. We train a discriminator that is not conditioned on the input domain, only on the NOCS maps, by using ground-truth NOCS samples only from simulation. In doing so we obtain a domain-agnostic discriminator able to inform our generator, irrespective of the image domain. We demonstrate that our method produces more coherent maps as compared to a supervised method on out-of-distribution data.

## 2 Related Work

In this work, we focus on deriving 2D-3D correspondences from RGB data alone. We train using semi-supervised methods for NOCS prediction. To do so we utilize GANs which allow us to adapt our model to a target domain without the need for labels from that target domain.

**Supervised NOCS**: To train the NOCS predictor, Wang et al. [4] proposed a fully supervised method whereby a convolutional head is trained using pixel-level loss between the predicted and ground-truth NOCS maps. To predict a NOCS map, Wang et al. [4] proposed adding a NOCS head to the MaskRCNN model [14], conditioning the head on the MaskRCNN-proposed regions of interest, and supervising it using ground truth NOCS. Labeled real data used to train this model is expensive to obtain because it requires determining the position and orientation of objects of interest relative to the camera. Because of the difficulty in obtaining real-world data they heavily rely on simulation. Nonetheless, real data was necessary to familiarize the model with that domain. We aim to avoid labeling real-world data while enabling our model to perform well in this domain.

**Self-supervised 3D perception**: Models trained on one domain (single style or source of data) struggle to generalize well, prompting work on self-supervised learning for 2D-3D correspondences and 3D understanding. Modern models have leveraged RGB and Depth to demonstrate domain adaptation through the use of self-ensemble techniques [9, 7]. Multi-modal RGB-D data also fueled methods that rely on shape priors and mask rendering [13], cycle consistency [? ], and 3D reconstruction [12] for training 3D Perception models. However, all these methods rely on the availability of accurate depth.

**Generative Adversarial Models**: Unique in approaching the training process from a game theoretic perspective, Generative Adversarial Models or GANs achieve their objective by pegging two agents, a generator and a discriminator, against each other [15]. A discriminator is tasked with distinguishing between ground-truth and predicted data, whereas the generator is given the task of deceiving it by generating more authentic predictions. This method of learning has been popular in the domain of image generation and styling [16] but it made its way into the field of domain adaptation mostly as an auxiliary tool for augmenting training data used by other models [17]. In this work, we leverage this style of training to directly supervise a model tasked with predicting NOCS maps. We use a discriminator, that is not conditioned on the input domain directly, allowing it to be impartial to it. The discriminator would thus be able to supervise the model irrespective of what domain its input is derived from.

# 3 Problem Statement

Let $I_i \in \mathbf{R}^{H \times W \times 3}$ be an RGB image such that $I_i \in \mathbf{I} = \mathbf{I_R} \cup \mathbf{I_S}$ where $I_R$ and $I_S$ are the sets of real and simulated images respectively. Let $N(I_i) \in \mathbf{N} = \mathbf{R}^{H \times W \times 3}$ be the ground truth NOCS map of $I_i$. The aim is to obtain a model $f_\Theta : \mathbf{I} \rightarrow \mathbf{N}$. Specifically, assuming that ground truth NOCS maps $N(I_i)$ are only defined for simulated images $I_i \in \mathbf{I_S}$ we want to better predict NOCS maps of real images $I_i \in \mathbf{I_R}$ making our objective:

$$\underset{\Theta}{\text{minimize}}(\|N(I_i) - \hat{T} f_\Theta(I_i)\|) \quad \forall \, I_i \in \mathbf{I_R} \tag{1}$$

Where $\hat{T}$ helps us avoid penalizing the rotation of symmetric objects along their axis of symmetry, making $\hat{T} = $ identity for non-symmetric objects and $\hat{T} = \text{argmin}_{T \in S_i} \|N(I_i) - \hat{T} f_\Theta(I_i)\|$ otherwise.

# 4 Method

The aim of this work is to mitigate the issues associated with sim-to-real domain shift by using *unlabeled* real-world data at training time. We observe that normalized 3D shapes of intra-category objects are similar in simulated and real data, thus their respective NOCS distributions $N(I_i \in \mathbf{I_S})$ and $N(I_j \in \mathbf{I_R})$ ought to be similar. This forms a key insight for our method.

We propose a model that incorporates two parts: a generator $f_\Theta : I_i \rightarrow \mathbf{N}$ that learns to predict NOCS from an image, and a discriminator $d_\Phi : N(I_i) \rightarrow [0, 1]$ that learns to differentiate predicted NOCS maps from ground truth ones. The discriminator learns only by observing simulated ground truth which we hypothesize to be sufficient for teaching the discriminator how to identify the proper shapes of normalized objects.

$$\begin{aligned} \underset{\Phi}{\text{minimize}} \quad & \frac{-1}{N} \sum_i \log\left(1 - d_\Phi(f_\Theta(I_i))\right) + \frac{-1}{M} \sum_j \log\left(d_\Phi(N(I_j))\right) \\ \text{subject to} \quad & I_i \in \mathbf{I}, \; i = 1, ... N \\ & I_j \in \mathbf{I_S}, \; j = 1, ... M \end{aligned} \tag{2}$$

In turn, a generator is trained to trick the discriminator into mistakenly labeling predicted NOCS maps as real. This yields the following objective for the generator:

$$\begin{aligned} \underset{\Theta}{\text{minimize}} \quad & \frac{-1}{N} \log(d_\Phi(f_\Theta(I_i))) \\ \text{subject to} \quad & I_i \in \mathbf{I}, \; i = 1, ... N \end{aligned} \tag{3}$$

Since our discriminator can differentiate generated NOCS from ground truth irrespective of the image source conditioning the generator, we can train the generator on both real and simulated data without the need for ground truth labels for real data. For every training batch, we first update the discriminator weights and then use the updated discriminator to supervise the generator. The discriminator loss can be incorporated with other supervised losses to mitigate the risk of mode-collapse.

# 5 Evaluation

**Dataset**: We use the dataset from Wang et al. [4], which consists of two training splits a simulated Context-Aware MixEd ReAlity (CAMERA) split and a REAL split containing seven scenes, and an evaluation split REAL275 containing six scenes. The dataset has six categories: bottle, bowl, camera, can, laptop, and mug. We only use the ground truth from the CAMERA split through training. Similar to [4], we use COCO data to augment our training. COCO and REAL images (not labels) are used in supervising our GANOCS model, whereas REAL275 is used for evaluation only.

**Setup**: Following prior works [4, 9] we base our model on MaskRCNN by adding our GANOCS generator and discriminator as a MaskRCNN head. The GANOCS generator is conditioned on the regions of interest proposed by MaskRCNN. This generator is composed of three separate networks, each of which is tasked with predicting one of the $(r, g, b)$ channels as continuous values. A weighted L2 loss is used to add further supervision over simulated data. The GANOCS discriminator is

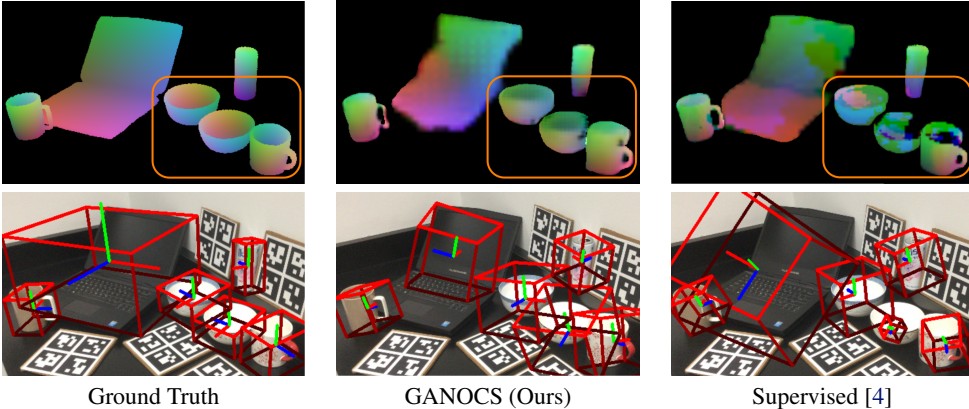

|  Ground Truth | GANOCS (Ours) | Supervised [4] |

Figure 2: Qualitative results on out-of-distribution evaluation data, demonstrating the ability of GANOCS to produce more stable NOCS maps as compared to the supervised baseline [4]. All models are trained without any real-data labels. The orange boxes on NOCS maps highlight areas of significant difference.

composed of a fully convolutional network with LeakyReLU activation and a final sigmoid action as suggested by Radford et al. [18]. We dedicate a 4 layer discriminator for every object category. We freeze the MaskRCNN backbone and train GANOCS on a single A5500 GPU for 200,000 steps with a batch size of 4. We use the work by Wang et al. [4] as a baseline and train it in a supervised manner using simulated data with batches of size 6.

**Preliminary Results**: Figure 2 shows our qualitative results demonstrating the ability of GANOCS to generate 2D-3D correspondence maps. Preliminary quantitative results show that our model can improve Mean-Average Precision (mAP) for $15°, 5cm$ and $15°, 10cm$ by 2%. Notably, our discriminator learns to adapt to symmetric objects without explicitly being supervised with symmetry-aware losses as is done by supervised methods [4]. To isolate the effectiveness of our discriminator, we train our model without any direct supervision. Figure 3

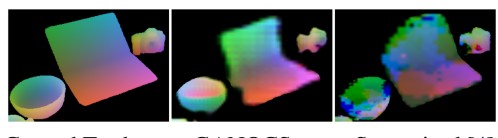

Ground Truth    GANOCS    Supervised [4]

Figure 3: Visual demonstrating results comparing GANOCS when trained with no direct supervision, highlighting the benefit of our method in avoiding artifacts on out-of-distribution data.

demonstrates that the supervised model shows inconsistency and non-smooth patches when used over out-of-distribution real-world data whereas our model, when using only the discriminator as loss, shows more consistent output.

## 6 Conclusion

We present GANOCS a method that trains an RGB-based NOCS predictor using a GAN style discriminator for the purpose of bridging the domain gap between simulation and real data without the need for labeling real data. We present qualitative results demonstrating the benefit of our work for bridging the sim-to-real gap.

Our future work will involve more extensive real-world evaluation and quantitative analysis in order to test the extent of our model's capabilities. We intend to explore the use of more recent NOCS architectures [8, 9] and improve training using GAN regularization and reconstruction techniques [19]. Our eventual aim of this work is to enable the use of NOCS representations for mobile manipulation by extending prior works [2, 3] from controlled tabletop settings to less structured large-scale spaces.

**Acknowledgments**

We thank the reviewers for their detailed feedback.

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
