# OpenReview forum: "GANOCS: Domain Adaptation of Normalized Object Coordinate Prediction Using Generative Adversarial Training"
_robot-learning.org/CoRL/2023/Workshop/OOD — OOD Workshop @ CoRL 2023_

### Official Review · Reviewer_wQyX · 2023-10-16
**Weak accept**

**Rating:** 5
**Confidence:** 4

**Review:**

This paper proposed a GAN based method to perform sim-to-real domain adaptation to a 2D-to-3D task predicting NOCS from RGB images. The proposed method trains with labeled simulated images and unlabeled real images and hopes to achieve indistinguishable output distributions between two different input distributions. This paper aligns well with the theme of this OOD workshop as it falls in the general category of sim-to-real domain adaptation problems. The quality of writing is clear and easy to follow, although the idea of using GAN for domain adaptation is not entirely new. The problem this paper tries to tackle is significant, and can be generalized further to other applications where real-world labels are much harder to obtain than simulated labels. The strength of this method is that it does not require OOD data labels in order to achieve this domain adaption. However, the biggest weakness of this method is the assumption that the output distributions on sim and real data are similar. In the general OOD cases, this may not be true. For example, if the inference image contain OOD geometries never seen during training, this method can potentially fail to generalize in those situations. The result of this paper is not very impressive. As seen from Figure 2, I cannot tell how GANOCS is qualitatively better than the baseline where no domain adaptation is considered in training. Figure 2 is also not very rigorous as the third column uses a slightly different image from the 1st and 2nd column. Similarly for Figure 3, GANOCS and the no-adaption baseline are different but equally inaccurate compared to the ground truth. It’s hard to say which one is better.

---

### Official Review · Reviewer_kd2g · 2023-10-16
**Applying GANs to enable generalization to domains where only unlabeled data is available is a promising strategy, but results are very limited here.**

**Rating:** 6
**Confidence:** 3

**Review:**

Summary:

This paper presents a method for achieving sim2real transfer (and more generally, domain transfer) for identifying 2d to 3d object correspondence for RGB data via NOCS prediction. The NOCS prediction task is notable in that acquiring labeled RGB / NOCS pairs is easy in simulation (where the 3d object models are known), but difficult in the real world. The authors propose a method to learn a predictor which generalizes to real world images using labeled simulation data and unlabeled real-world data. Specifically, the method uses a GAN style discriminator which aims to distinguish GT NOCS from predicted NOCS. This discriminator is trained only on simulated data (where GT NOCS are available), but provides supervision to the NOCS predictor (generator) on real-world inputs as well. The results show qualitative improvement in NOCS quality on real-world inputs relative to supervised training alone.

Review:

Strengths:

- The paper is well written, motivates the problem well and explains the approach clearly.
- The problem of sim2real transfer is important and a key source of distribution shift when deploying models in domains where simulated data is cheap but real-world data is expensive. Moreover, the approach proposed of using a discriminator trained on labeled data to provide supervision on unlabeled data that is more representative of real-world deployment is promising and could be generalized to other domains.

Weaknesses:

- Results are not immediately compelling -- in Figure 2, while the predicted NOCS visually look smoother and have less artifacts, it isn't clear whether this translates to more accurate pose estimation. Future quantitative analysis investigating whether these qualitative improvements translate to benefits for the downstream pose estimation task.
- The key insight in this work is that the predicted NOCS for real-world images and synthetic images are similar. In particular, the assumption is that the discriminator trained only on synthetic data can generalize better to the real-world inputs than a predictor directly trained on synthetic data. I would imagine this relies on ensuring the distribution of real-world GT NOCS matches the distribution of synthetic GT NOCS (e.g. having the same distribution over object classes). The paper would be strengthened with a deeper exploration of this requirement, potentially highlighting the limits of such an approach in cases where this doesn't hold (e.g. real-world objects that didn't cleanly correspond any synthetic counterparts).
- Training GANs can be difficult due to issues of mode collapse or imbalance between the generator and discriminator. How would you tune parameters without access to real-world labeled data to test the generalization capabilities?

---

### Decision · Program_Chairs · 2023-10-17

**Decision:**

Accept

**Comment:**

We agree with the reviewers’ assessment that this work is technically sound and will contribute to productive, topical discussions at the 2023 Workshop on OOD Generalization in Robotics. In particular, we appreciate that this work tackles directly a problem in OOD generalization towards real-world robot deployment. We recommend the authors incorporate the reviewers’ feedback into their camera-ready submission to further improve their manuscript.